# Hydrothermal Transformation of Eggshell Calcium Carbonate into Apatite Micro-Nanoparticles: Cytocompatibility and Osteoinductive Properties

**DOI:** 10.3390/nano13162299

**Published:** 2023-08-10

**Authors:** Adriana Torres-Mansilla, Pedro Álvarez-Lloret, Raquel Fernández-Penas, Annarita D’Urso, Paula Alejandra Baldión, Francesca Oltolina, Antonia Follenzi, Jaime Gómez-Morales

**Affiliations:** 1Departament of Geology, University of Oviedo, 33005 Oviedo, Spain; uo272371@uniovi.es; 2Laboratory of Crystallographic Studies, IACT-CSIC-University of Granada, Avda. Las Palmeras, n° 4, 18100 Armilla, Spain; raquel.fernandez@csic.es; 3Dipartimento di Scienze della Salute, Università del Piemonte Orientale, “A. Avogadro” Via Solaroli, 17, 28100 Novara, Italy; annarita.durso@uniupo.it (A.D.); francesca.oltolina@med.uniupo.it (F.O.); antonia.follenzi@med.uniupo.it (A.F.); 4Departament of Oral Health, Faculty of Dentistry, Universidad Nacional de Colombia, Bogotá 111321, Colombia; pabaldione@unal.edu.co

**Keywords:** calcite, apatite, organic matrix, hydrothermal conversion, cytocompatibility, osteogenic differentiation

## Abstract

The eggshell is a biomineral consisting of CaCO_3_ in the form of calcite phase and a pervading organic matrix (1–3.5 wt.%). Transforming eggshell calcite particles into calcium phosphate (apatite) micro-nanoparticles opens the door to repurposing the eggshell waste as materials with potential biomedical applications, fulfilling the principles of the circular economy. Previous methods to obtain these particles consisted mainly of two steps, the first one involving the calcination of the eggshell. In this research, direct transformation by a one-pot hydrothermal method ranging from 100–200 °C was studied, using suspensions with a stoichiometric P/CaCO_3_ ratio, K_2_HPO_4_ as P reagent, and eggshells particles (Ø < 50 μm) both untreated and treated with NaClO to remove surface organic matter. In the untreated group, the complete conversion was achieved at 160 °C, and most particles displayed a hexagonal plate morphology, eventually with a central hole. In the treated group, this replacement occurred at 180 °C, yielding granular (spherulitic) apatite nanoparticles. The eggshell particles and apatite micro-nanoparticles were cytocompatible when incubated with MG-63 human osteosarcoma cells and m17.ASC murine mesenchymal stem cells and promoted the osteogenic differentiation of m17.ASC cells. The study results are useful for designing and fabricating biocompatible microstructured materials with osteoinductive properties for applications in bone tissue engineering and dentistry.

## 1. Introduction

The eggshell (ES) of hens is a complex biomineral structure that functions as a protective barrier for the egg content, allowing the development of the chick embryo out of the uterus [1]. It is composed of a calcified shell attached to the outer surface of a polymeric membrane, known as the eggshell membrane (ESM) [2].

The shell is composed of ~96 wt.% calcium carbonate (CaCO_3_) in the form of calcite, an intracrystalline occluded organic matrix (OM, 1–3.5 wt.%), and the remaining components include Mg, P, H_2_O, and other trace elements [2]. The OM is composed of proteins, glycoproteins, and proteoglycans. Among these organic components, it is worth mentioning egg white proteins such as ovalbumin, lysozyme and ovotransferrin, ubiquitous proteins (e.g., osteopontin and clusterin), and organic constituents unique to the process of shell calcification. This latter group comprises dermatan and keratan proteoglycans, ovocleidins and ovocalyxins [1]. A mass spectrometry-based high-throughput proteomic study of the acid-soluble OM of the chicken eggshell identified more than 500 proteins in the shell [3]. This OM content is fundamental in eggshell calcification and formation, leading to a highly structured material with intrinsic porosity. The organic components have been proven to control the nucleation and growth of calcium carbonate during in vitro experiments, yielding crystalline morphologies similar to those of the shell formed in vivo [4,5]. 

ES is an abundant source of naturally occurring carbonate formed from biologically controlled biomineralization processes. However, these materials and their associated ESM represent waste by-products discarded in the food industry. This waste is commonly considered useless and is often disposed of in landfills without any pretreatment. This waste management is not a desirable practice in view of the environmental odor from biodegradation. The disposal is ecologically harmful and implies the loss of potentially valuable biomaterials [6,7]. According to estimates [8], the amount of ES discarded in 2018 was close to 8.5 million tons and represented the fifteenth largest source of pollution in the food industry. Reusing this by-product would reduce the economic and environmental burden related to its processing. Different proposals to transform ES waste into valuable materials include its use as a low-cost adsorbent for the removal of ionic pollutants from the aqueous solution [9,10], as a solid base catalyst used for biodiesel pollutant minimization [11], as a dietary calcium supplement in human and animal food, or plants fertilizer [12], or as an environmentally friendly raw material to produce calcium phosphate apatite (Ap) bioceramics [13]. 

Apatites are the main inorganic component of bones and teeth in mammals. Synthetic nanocrystalline apatites (Ap NPs) exhibit high biocompatibility, bioactivity, and a relatively high bioresorbability in biological media. Their synthesis and characterization methods can be approached from many different perspectives and offer numerous technological and industrial applications [14,15]. The modulation of their physicochemical properties has allowed the use of Ap NPs in advanced biomedical applications, such as bone tissue regeneration [16], as nanocarriers for drug delivery [17], or as luminescent probes in bioimaging [18,19,20], among others. In the field of bone tissue regeneration, the use of other inorganic nanoparticles, for example, platinum nanoparticles (Pt NPs), has been reported to enhance the osteogenic differentiation of human dental follicle stem cells [21]. However, particles based on Ag NPs present advantages in terms of compositional compatibility with calcified tissues.

ES-derived Ap particles have been obtained mainly by two-step processes involving the calcination of powdered ES to CaO and CO_2_ at temperatures above 800–900 °C, followed by a reaction of CaO with phosphoric acid [22,23,24] or with tricalcium phosphate and (NH_4_)_2_HPO_4_ in hydrothermal conditions at high temperatures [25,26]. These multi-step processes involving a calcination step fully degrade or destroy the intracrystalline OM and any surface organic residue of the ES powder. Additionally, these processes are energetically expensive and environmentally poorly sustainable. 

Recently, a “one-pot” method to directly transform biogenic CaCO_3_ particles into apatite micro-nanoparticles under mild hydrothermal conditions was set up [27]. The method avoids the calcination step as well as the use of acids and other additives. In processes of mineral replacement from biogenic calcium carbonate under mild conditions, the organic matter content can play a critical role. Bleaching with sodium hypochlorite (NaClO) is a common practice to remove the surface organic/tissue residues of biominerals [28]. The treatment must not affect the intracrystalline occluded OM. 

In this research, we investigated the transformation of bleached (NaClO-treated) powdered ES into Ap micro/nanoparticles with cytocompatible and osteoinductive properties for osteogenic applications using the one-pot hydrothermal method. Additionally, we investigated the transformation of non-bleached powdered ES (NaClO-untreated) into Ap to gain insight into the role of the surface organic residues in the transformation process as well as on the physicochemical and morphological characteristics of the obtained Ap particles. The aim is to contribute to developing a strategy for reusing this biogenic waste in particles of biomedical interest by setting up a cost-effective (with a reduced number of processing operations) and eco-friendly procedure by means of a low-temperature hydrothermal method.

## 2. Experimental Section

### 2.1. Materials and Methods 

The eggshells were collected from hen eggs. The ESM was carefully removed and discarded. The remaining CaCO_3_ shells were crushed and divided into two groups. The first group was treated with NaClO 5% *v*/*v* for 24 h, washed with tap water, dried in air, and labeled ES(t); the second one was not bleached; it was only washed with tap water and air-dried. This untreated group was labeled ES(u). Both ES groups were ground in an agate mortar and sieved to particle sizes Ø < 50 μm. 

The experiments were performed inside a Memmert UNB 200 oven (Memmert, Schwabach, Germany) with circulating forced air, using a specially designed aluminum box that contained four 10 mL PTFE tubes. The box had an aluminum cap coated with PTFE to close the ensemble. The tubes were filled to 70% with suspensions of either ES(t) or ES(u) and K_2_HPO_4_ in stoichiometric K_2_HPO_4_/CaCO_3_ ratios with respect to hydroxyapatite (0.6). All suspensions were prepared with ultrapure Milli-Q water (resistivity 18.2 M.Ω.cm, Millipore, Merck, Burlington, MA, USA) and analytical grade reactants (≥97–99%; Sigma Aldrich, Darmstadt, Germany). Hydrothermal reactions were carried out between 100 and 200 °C (Δ 20 °C) for 7 days. Then, the precipitates were washed by centrifugation with deionized water (3 cycles, 9000 rpm each) to remove unreacted species and freeze-dried under vacuum (3 millibars) at −50 °C overnight. 

### 2.2. Physicochemical Characterization

The physicochemical characterization was performed by X-ray diffraction (XRD), high-resolution scanning electron microscopy (HRSEM), energy dispersive X-ray spectroscopy microanalysis (EDX), Fourier transform infrared-attenuated total reflectance (FTIR-ATR) and Raman spectroscopies, thermogravimetry (TGA), dynamic light scattering (DLS) and ζ-potential against pH. 

XRD data were obtained using a PHILIPS X’PERT PRO X-ray diffractometer (Almelo, The Netherlands). For diffraction experiments, the working conditions were Cu Kα_1_ radiation (λ = 1.54059 Å) at 45 kV and 40 mA. XRD patterns were collected from 4° to 60° (2θ), with a 2θ scan step of 0.007° and a counting time of 1 s per step. Phase identification was performed by matching the experimental diffraction patterns with those included in the crystal structure information provided by the American Mineralogist Crystal Structure Database (AMCSD). The average crystallite sizes were determined as Lvol-IB values (i.e., volume-weighted mean crystallite sizes calculated by FWHM and integral amplitude) using TOPAS V.4.2 software.

High-resolution scanning electron microscopy was performed with an AURIGA FIB-FESEM microscope (Carl Zeiss SMT Inc., Danvers, MA, USA) with an accelerating voltage of 5 kV, coupled to an X-ray energy-dispersive spectrometer from Oxford Instruments for chemical micro-analysis. The samples were carbon-coated prior to observation. 

FTIR-ATR characterization was performed with an Invenio R FTIR spectrometer (Bruker, Billerica, MA, USA) equipped with an attenuated total reflectance (ATR) accessory of a diamond crystal. Spectra were recorded within the wavenumber range from 4000 cm^−1^ to 400 cm^−1^ at a resolution of 2 cm^−1^. Raman spectra were recorded with a LabRAMHR spectrometer (Jobin–Yvon, Horiba, Kyoto, Japan). The excitation line was provided by a diode laser emitting at a wavelength of 532 nm. A Peltier-cooled charge-coupled device (CCD) (1064 × 256 pixels) was used as the detector.

Crystal size distributions (CSD, mass) and ζ-potential were measured with a Malvern Zetasizer Nano ZS analyzer (Malvern Instruments Ltd., Malvern, UK) using, respectively, quartz and disposable polystyrene cuvettes filled with aqueous suspensions of ~0.5 mg/mL. For ζ-potential versus pH measurements, we used 0.1 M HCl and 0.1 M NaOH as titrant solutions without any additional electrolytes. 

Thermogravimetric analyses (TGA-DSC3+, Mettler Toledo, Columbus, OH, USA) were performed under air between 30 and 950 °C at a constant 20 °C /min heating rate. The mass loss during heating was used to determine the water (up to 180 °C), organic matter (range 180 to 550 °C), and mineral (above 550 °C) contents, within which the fraction of carbonate component (between 55 to 850 °C) was also calculated.

### 2.3. Biological Tests

#### 2.3.1. Cell Cultures

The human osteosarcoma cell line MG-63 (ATCC^®^CRL–1427™) was maintained in Dulbecco modified Eagle’s medium (DMEM) (Sigma-Aldrich, Milan, Italy) supplemented with 10% fetal bovine serum (FBS), antibiotic solution (streptomycin 100 µg/mL and penicillin 100 U/mL, Sigma-Aldrich, Milan, Italy) and 2 mM L-glutamine (complete medium). M17.ASC cells (a spontaneously immortalized mouse mesenchymal stem cell clone from subcutaneous adipose tissue) [29] were cultured in Claycomb medium (Sigma-Aldrich, Milan, Italy), supplemented as described above. Cells were incubated at 37 °C in a humidified atmosphere with 5% CO_2_, and they were regularly split when sub-confluent at a ratio of 1:8 and 1:10 for MG-63 and m17.1ASC, respectively.

#### 2.3.2. Cytocompatibility

MG-63 cells (5 × 10^3^/well) and m17.ASC (2 × 10^3^/well) were seeded in 96-well plates, and 24 h after, different concentrations (ranging from 0.1 to 100 µg/mL) of particles were added in 100 µL of fresh complete medium. The particles used in this study were ES(u), ES(t), Ap(t)-160 °C and Ap(t)-200 °C. After 72 h of incubation, cell viability was evaluated by the 3-(4,5-Dimethylthiazol-2-yl)-2,5-diphenyltetrazolium bromide) (MTT, Sigma) colorimetric assay as described in Cano Plá et al. [19]. Briefly, 20 μL of MTT solution (5 mg/mL) in phosphate buffer saline (PBS) was added to each well, and the plate was incubated at 37 °C for 2 h. Afterward, supernatants were carefully aspirated, and then 125 μL of 0.2 N HCl in isopropanol was added to dissolve formazan crystals. Then, 100 μL were removed, and the optical density was measured in a multi-well reader (2030 Multilabel Reader Victor TM X4, Perkin Elmer, Waltham, MA, USA) at 570 nm. Hydrogen peroxide (1 μM) was used as control of toxicity. The absorbance value of untreated cells was taken as 100% viability, and values obtained from cells undergoing the different treatments were compared to this value. Experiments were performed at least three times using 3 replicates for each sample.

#### 2.3.3. Osteogenic Differentiation of m17.ASC

m17.ASC cells were seeded onto 12-well plates at a density of 1 × 10^4^ cells per well. After 24 h, they were treated for 14 days with a concentration of 25 μg/mL of Es(u) and apatite NPs transformed at 200 °C to evaluate the osteoinductive potential effects of NPs. Cells were compared to a positive control group that was cultured for the same time with an osteogenic medium (Ob) containing DMEM, FBS 10%, 50 μg/mL ascorbic acid (MW 176.12), 10 mM β-glycerophosphate (MW 216.04), and 10 nM dexamethasone (Sigma-Aldrich). The culture medium was replaced every 3 days during the treatment period, as previously described in [29].

#### 2.3.4. Alkaline Phosphatase (ALP) Staining and Quantitative Analysis

The osteogenic differentiation was estimated by alkaline phosphatase (ALP) staining and quantified with ImageJ 1.48v software as described in Dupont et al. [30]. Briefly, after 14 days of culture with the particles, cells were washed three times with PBS, fixed with 4% paraformaldehyde (4% PFA; Sigma-Aldrich) for 15 min, and stained with an alkaline phosphatase detection kit (Millipore, Merck Millipore, Milan, Italy) according to the manufacturer’s protocol. Untreated cells and cells differentiated with the osteogenic medium were used as a negative and positive control, respectively. Images were acquired under optical microscopy at 200× magnification. For ALP quantification, the intensity of violet staining was estimated as the integrated density (INT.DEN.) by ImageJ analysis. This value was normalized to the number of cells for each picture and was expressed as an arbitrary unit (A.U.). Experiments were performed at least three times.

#### 2.3.5. Alizarin Red Staining (ARS) and Quantitative Analysis

To determine the calcium deposition in m17.ASC cells, as a consequence of the osteogenic differentiation after a 14-day treatment with osteogenic medium or particles, Alizarin Red S staining was carried out as described in Sutthavas et al. [31]. Briefly, cells were washed with PBS pH 7.2, fixed with 2% PFA (2 wt% in PBS), and then stained with Alizarin Red S solution (40 mM), pH 4.1, for 30 min at room temperature. Afterward, the cells were washed thrice with bi-distilled water to remove the non-specific precipitation, and then samples were analyzed and photographed by optical microscopy at 200×. To quantify the staining, mineralized deposits were dissolved in 10% acetic acid (Sigma-Aldrich) for 30 min, and then 150 μL of each sample was collected in a 96-well plate to measure their optical density in a multi-well reader (2030 Multilabel Reader Victor TM X4, Perkin Elmer) at 405 nm. Experiments were performed at least three times using 3 replicates for each sample.

#### 2.3.6. Statistical Analysis

Data were statistically analyzed and are expressed as mean ± standard deviation of at least three replicates. Statistical analyses were performed using one-way ANOVA with Bonferroni’s post-test for grouped analyses using GraphPad Prism version 7.0 for Mac, GraphPad Software (GraphPad Prism, San Diego, CA, USA). Statistical differences between the treatments were considered significant when p values were *p* < 0.05 (*), *p* < 0.01 (**), *p* < 0.001 (***), and *p* < 0.0001 (****). 

## 3. Results

### 3.1. Crystallographic, Morphological, Compositional, and Spectroscopic Features

In both ES groups, the starting material (Figure 1, bottom) was identified as calcite, depicted by its characteristic XRD pattern (PDF 00-005-0586). The diffractograms showed the most intense reflection at 2θ = 29.3° (plane 104). The samples displayed a nanogranular texture characteristic of biological calcite [32]. Treatment with NaClO and milling neither induced the formation of additional crystalline phases nor seemed to modify the particle morphology. 

After hydrothermal reaction (Figure 1a) at 100 °C, the ES(t) group showed the partial transformation from ES(t) into Ap(t), characterized by the emerged reflections at 2θ = 25.87° (002) and those at 31.77° (211), 32.19° (112), 32.90° (300), and 33.97° (202), respectively (PDF 01-1008). The total transformation from ES(t) into Ap(t) occurred at 180 and 200 °C, witnessed by the lack of the (104) calcite reflection in these diffractograms. The average crystallite size determined by XRD (i.e., the crystalline domain size) of both ES(t) and Ap(t)-200 °C were 190 and 80 nm, respectively. SEM pictures of Ap samples showed crystals with globular morphologies forming larger aggregates (Figure 1(ai,aii)), similar to those found by other authors [33]. The EDX microanalysis of the Ap(t)-200 °C sample yielded 39.30 wt.% Ca, 17.85 wt.% P, 39.8 wt.% O, 2.65 wt.% K, and 0.65 wt.% Mg, with (Ca+Mg)/P = 1.74. Note that in this analysis, the C was ruled out since we used C to metalize the samples prior to SEM observation, but it did not influence the calculation of the (Ca+Mg)/P molar ratio. At intermediate temperatures (i.e., 100–160 °C), the SEM images showed globular morphologies due to both phases ES(t) and Ap(t) being indistinguishable (Figure 1(aiii,aiv)). 

In parallel, in the ES(u) group, the sample submitted to hydrothermal conversion at 100 °C did not undergo transformation (Figure 1b), but at 120 and 140 °C, the transformation was very high, and at 160–200 °C, the ES(u) sample transformed completely. The EDX composition of the Ap(u)-200 °C sample consisted of 38.9 wt.% Ca, 18.0 wt.% P, 39.9 wt.% O, 1.5 wt.% K, and 0.5 wt.% Mg, yielding a (Ca+Mg)/P mol ratio of 1.67. The average crystallite sizes of both ES(u) and Ap(u)-200 °C were 19 nm and 55 nm, respectively. The larger crystallite size of ES(t) compared to ES(u) revealed the growth of the CaCO_3_ crystalline domain size when removing the surface OM of the sample. Besides this observation, the notable finding was that many apatite single crystals displayed a hexagonal plate morphology, whose width and height were up to 1.5 µm and 0.75 µm (Figure 1(bii)), respectively, and sometimes showed a centric hole (Figure 1(biii)). These morphologies strongly differed from that obtained in the ES(t) series. Figure 2b–d show the high-resolution images of these remarkable morphologies obtained at 120, 160 and 200 °C, compared with the spherulitic one obtained in the ES(t) series (Figure 2a). The finding highlights the strong impact of the surface OM of the ES(u) on the mineral replacement reaction and on the morphological aspect of the Ap particles. At 200 °C, however, the morphology became more granular, as in the homologous experiment with the ES(t) group. The average particle sizes for the Ap(u)-200 °C and Ap(t)-200 °C samples measured from SEM images were 87.5 ± 13.8 and 131.3 ± 10.6 nm, respectively. 

The thermogravimetric characterization of ES(u), ES(t), Ap(u)-200 °C and Ap(t)-200 °C (Figure 3a,b) showed that the NaClO treatment removed approximately 50% of the OM content of ES(t) with respect to the ES(u) sample, most likely all surface organic components. Both samples had different mineral content, approximately 98% and 95%. On the other hand, the transformed samples, Ap(u)-200 °C and Ap(t)-200 °C, showed a slight reduction in OM content and a significant removal of the mineral carbonate (~2–5%) component (see Table 1).

The ATR-FTIR spectra also evidenced the mineral conversion (Figure 4). Both ES(t) and ES(u) particles (Figure 4a,c, bottom) presented the calcite CO_3_ vibrations ν_4_ (asymmetric stretching) at 712 cm^−1^, at ~873 cm^−1^ the ν_2_ out-of-plane bending, and at ~1410 cm^−1^ the ν_3_ in-plane bending [34,35,36]. The ~1089 cm^−1^ peak (symmetric stretching, ν_1_) was not pronounced or identified in the figure. The peak position ν_3_ shifted to higher wavenumbers with respect to that of geogenic calcite (see Appendix A) because of the presence of Mg [37]. The 1088 cm^−1^ (ν_1_CO_3_) was, however, active in Raman, being the most intense to identify the calcite of ES (Figure 4b,d, bottom). Besides these peaks, the FTIR spectrum of ES(u) also showed a tiny band at 1650 cm^−1^, even smaller in ES(t), which corresponded to the amide II of side chains of proteins. 

When submitting samples to hydrothermal conversion, in both groups (Figure 4a,c) appeared the characteristic vibrational modes of PO_4_ groups of the apatite, i.e., at ~1020 cm^−1^, the asymmetric stretching mode (ν_3_); at ~960 cm^−1^ the symmetric stretching (ν_1_), at ~600 and ~560 cm^−1^, the bending modes (ν_4_), and at ~470 cm^−1^ , the ν_2_PO_4_ [38]. In addition, the vibrational modes ν_2_CO_3_ at around 873 cm^−1^ and ν_3_CO_3_ at about 1410 cm^−1^ remained in the transformed samples, though with very low intensity, indicating that the apatites were CO_3_^2—^substituted. In this case, the ν_3_CO_3_ appeared to split into two bands, at ~1410 and ~1460 cm^−1^, as usually observed in CO_3_^2—^substituted apatites [20]. A deconvolution was performed on the low-intensity absorption band centered at 873 cm^−1^ (Appendix A), revealing three different sub-bands located at ~880, ~873, and ~870 cm^−1^ attributed to A-type (CO_3_^2—^replacing OH^-^), B-type (CO_3_^2—^ replacing PO_4_^3-^) and labile CO_3_^2—^species located at the surface of the nanoparticles [38,39]. The degree of carbonation was also estimated using a quantification methodology [38] that compared the relative intensity of the FTIR ν_2_CO_3_ carbonate bands relative to the ν_1_- ν_3_PO_4_ phosphate bands. Following this methodological approach, the overall carbonation degree was close to 5 wt.% for Ap(t) and 3.5 wt.% for Ap(u) obtained at 200 °C, in the range of that calculated by TG. Finally, the Raman spectra (Figure 4b,d) also showed the presence of the most intense mode ν_1_PO_4_ at 960 cm^−1^ and the ν_3_ at 1070 cm^−1^, while the ν_1_CO_3_ at 1088 cm^−1^ progressively disappeared when increasing the temperature. 

### 3.2. Crystal Size Distribution and ζ-Potential Versus pH of ES and Derived Ap Particles

The crystal size distribution and ζ-potential of particulate suspensions at pHs of physiological interest (i.e., 7.4 simulating the pH in blood [40], 7.2–7.4 in bone tissue [41], and 5.6–7 in a tumor microenvironment or inflamed tissue [42,43] were essential characteristics to evaluate the possible biomedical applications of ES and derived Ap micro-nanoparticles as nanocarriers for drug delivery, or as implantable materials in bone tissue regeneration.

Both ES(u) and ES(t) were composed of aggregates of nanoparticles showing a multimodal distribution (Figure 5a). The CSD of ES(u) showed particle sizes of around 50 nm and aggregates of up to 5.5 µm, while ES(t) was composed of particles with a median size of 68 nm and aggregates of 615 nm. The plot as cumulative volume-based distribution (Figure 5b) revealed the percentiles D_10_, D_50,_ and D_90_ of the distribution. In these samples, D_10_ was 44 and 88 nm, respectively, close to the individual particle size, while D_50_ (the median of the distribution) and D_90_ were affected by aggregation. On the other hand, Ap(u)-200 °C and Ap(t)-200 °C showed D_10_ values of 116 and 202 nm and D_50_ values of 574 and 742 nm, respectively, reflecting the presence of short-range aggregates in both samples. In these distributions, D_90_ was affected by long-range aggregation. 

The evolution of ζ-potential against pH reveals that values of ES(t) and Ap(t)-200 °C were negative and slightly lower than those of ES(u) and Ap(u)-200 °C at pHs of physiological interest (Figure 5c,d). This finding reflected the presence of negative surface charges, but the low values of the ζ-potential (below −10 mV) indicated that particles tend to aggregate rather than to disperse in aqueous suspensions, except at high pH. According to these results, both ES and derived Ap particles could be more useful for applications as implantable materials in bone regeneration. 

### 3.3. Biological Tests

#### 3.3.1. Cytocompatibility

The MTT assay was used to test the cytocompatibility of the ES(u), ES(t), Ap(t)-160 °C, and Ap(t)-200 °C particles on both MG-63 cells and m17.ASCs cells. In particular, cells were incubated for 3 days with different particle concentrations ranging from 100 to 0.1 μg/mL. As shown in Figure 6a, all tested particles were cytocompatible on MG-63 cells since, in all cases, no significant toxicity was observed. Similar results were obtained when m17.ASC cells were incubated with the lowest doses of both ES(u) and Ap(t) (Figure 6b). Conversely, the viability of m17.ASC cells resulted in being about 90% when the highest concentrations of the different types of particles were tested. In all cases, none of the samples reached the toxicity of hydrogen peroxide, used as a positive control. 

#### 3.3.2. Effect of Eggshell and Apatite Particles on the Osteogenesis and Mineralization of m17.ASC cells

To evaluate the ability of ES and Ap particles to affect the osteogenic differentiation of mesenchymal stem cells, m17.ASC cells were incubated with 25 μg/mL (minimum effective dose reported in the literature [44,45]) of particles for 14 days in the absence of osteogenic stimulators, and then we evaluated by the staining assay the enzymatic ALP activity, a hallmark of the early stage of osteogenesis. For these experiments, we chose the ES(u) and Ap(t)-200 °C samples. In addition, m17.ASC cells cultured in an osteogenic medium and in a complete medium (untreated) were used as positive and negative controls, respectively. Our findings showed that the application of ES(u) particles to mesenchymal cells induced an increase in ALP production after 14 days, in contrast to untreated cells. Remarkably, an increase in violet signal, typical of ALP staining, was visualized when cells were incubated with the Ap(t)-200 °C particles (Figure 6c). In fact, the quantitative analysis (Figure 6d) showed that the intensity of violet staining associated with the particles (28531 and 31880 A.U. for ES(u) and Ap(t)-200 °C, respectively) was nearly comparable to the signal detected (38330 A.U.) in the case of differentiated cells with osteogenic medium. Thus, these findings suggest that the tested particles could play a role in early stages of the osteogenesis process.

Next, we investigated whether these nanoparticles could promote in m17.ASC cells the mineralization process, the final stage of osteogenesis characterized by the calcium salt deposits forming the bone matrix [46]. To examine the effect of these particles on the matrix mineralization, we carried out alizarin red staining after m17.ASC cells were treated with ES(u) and Ap(t)-200 °C particles as described above. As shown in Figure 6e, cells treated with both types of particles showed a behavior similar to that of the stem mesenchymal cells differentiated with osteogenic factors used as a positive control. In detail, the differentiated cells were surrounded by notable calcium precipitates visualized as a red nodule-like staining, which is a hallmark of the formation of mineralized matrix. These data were also supported by the quantitative analysis (Figure 6f), in which the intensity of ARS extracted from the stained wells was measured and expressed as the optical density (O.D.); in fact, in the presence of ES(u) and Ap(t)-200 °C, the mineralized matrix depositions in cells were nearly duplicated compared to the untreated group (negative control), confirming their role in mineralization.

## 4. Discussion

Most of the previous methods to obtain Ap particles from ground ES consisted of two steps, the first one being calcination (>800–900 °C), thus completely destroying the ES organic matrix. Herein, we applied the “one-pot” hydrothermal method [27]. The process can be classified as a mineral replacement reaction in which a Ca-salt (e.g., calcium carbonate) is replaced with another one having a lower solubility product (e.g., calcium phosphate apatite). It takes place by a dissolution–reprecipitation mechanism when the P/CaCO_3_ mol ratio (0.6 or higher), P reagent, and temperatures are properly applied. In such processes, the fluids involved dissolve the parent mineral phase and create an interfacial boundary layer. Depending on the composition of the fluid, this layer becomes supersaturated with one or more new phases of a different mineral type. One of these phases nucleates and starts an autocatalytic reaction, precipitating a new phase [47].

Bleaching of biominerals by immersion in a NaClO solution is a common practice to remove surface organic matter and tissue debris prior to material observation and analysis. This soft cleaning process normally preserves the intracrystalline OM. One objective of this research is to gain insights into the influence of these surface organic residues on the mineral replacement reaction. We have found differences in the minimum temperature to achieve the full conversion as well as in the morphology of the obtained Ap. Thus, when using ES(t), the temperature was 180 °C, while in the ES(u) group, it decreased to 160 °C, with a narrower temperature interval in which ES(u) and Ap(u) coexisted (from 120–160 °C). This finding highlights the catalytic effect of the OM in the transformation. In addition, most apatite crystals showed hexagonal plates, and some of them displayed a central hole, unlike those obtained in the ES(t) group at the same temperatures, whose morphologies were always spherulitic. These differences are presumed to be attributable to the different OM contents in ES(u) and ES(t), as demonstrated by thermogravimetry (Figure 2) and by the identification of the amide II of proteins in the FTIR-ATR spectrum (Figure 3). The treatment of ES with NaClO has been proven to remove almost half of the OM (that we presume present at the surface level of the particle) but preserves most of the intracrystalline OM. 

The mechanisms by which the OM affects the formation of hexagonal plates can be approached by considering that the apatite crystal structure belongs to the hexagonal system, space group P63/m, with a = b = 0.9418 nm and c = 0.6884 nm, and α = β = 90°, γ = 120°. The crystal morphology is dominated by the (010) and (001) surfaces exposing different terminations for interaction with water and biomolecules. The (001) surface is terminated by hydroxyl ions perpendicular to the {001} crystal plane, while the (010) can be phosphate-exposed, calcium-exposed, and hydroxyl-exposed [48] Our hypothesis is that due to the amount and variety of biomolecules in the ES(u) sample, after ES(u) dissolution, they remain available in the solution and influence the growth of apatite by their adsorption on both surfaces, thus being expressed in the growth morphology. The other interesting finding is the formation of hollow apatite crystals. To our knowledge, there are no references to this striking morphology in apatite, but we hypothesize that the presence of biomolecules in the granular calcite surfaces of the ES(u) can eventually induce the surface nucleation and growth of the apatitic phase, thus wrapping the granules. The further dissolution of the parent phase (whose solubility is lower) will release the intracrystalline macromolecules while the hole is created. Therefore, the free macromolecules influence crystal formation as before, expressing both surfaces in the growth morphology. A separate study of the formation of both morphologies in detail would be worth carrying out to confirm this hypothesis.

The ES OM has also been overlooked when employing the hydrothermal treatment in two-step transformation processes since a first step, either of calcination or chemical dissolution with HCl, was usually performed [33,49,50]. Besides the use of the one pot-methodology, the second part of the study consisted of the analysis of the biological properties of ES calcium carbonate and derived Ap particles. The cytocompatibility of each xenobiotic agent should be one of the first parameters to ascertain before testing their eventual biomedical application. Thus, we confirmed the non-toxicity of the different types of ES and Ap-derived particles by analyzing the viability of MG-63 and m17.ASC cells after their exposure to the particles. Our findings highlight that the mesenchymal stem cells were more sensitive to the inorganic particles than the osteosarcoma cell line. Nevertheless, it must be noted that the cell viability in all cases was always higher than 80%, which is above the cut-off of 70% indicated by ISO 10993–5:2009 [51]. Other calcium phosphates, such as amorphous calcium phosphate (ACP) particles, have been successfully synthesized from eggshell through a low-cost guided aggregation technique [52]. These ACP-eggshell derived nanoparticles presented higher resistance to recrystallization, nanostructural control, and biosafety compared to synthetic ACP particles. Taken together, these data confirm the high cytocompatibility of these calcium-based nanoparticles derived from eggshell, as described above. There is mounting evidence that calcium-based nanoplatforms are promising materials for bone regeneration applications [53].

Then, we analyzed the effect of these particles obtained by the hydrothermal transformation of ES on the osteogenic differentiation of the mesenchymal cell line m17.ASC. Particle-induced osteogenic cell differentiation has been linked to the presence of Ca^2+^ ions, which function as a potential osteoinductive trigger in mesenchymal cells, as previously established in human bone marrow-derived mesenchymal stromal cells (hBMSC) [54,55]. Ca^2+^ participates as a second messenger for the activation of the intracellular pathways, since Ca-mediated signals couple the membrane events generated by extracellular signals with the cytoplasmic biochemical cascades and nuclear gene expression programs required for cell differentiation. Furthermore, Ca^2+^ promotes the expression of proteins, including bone morphogenetic protein 2 (BMP-2), which participates in the osteogenic differentiation of hBMSCs [56], thus suggesting that a similar behavior in m17.ASC cells is likely. In particular, we assessed the ability of both ES (u) and Ap (t)-200 °C to affect the enzymatic ALP activity and the extracellular calcium depositions. ALP plays a critical role during the early stage of osteogenesis and generally reaches the maximum level at 2 weeks, while extracellular calcium nodules are a unique feature of the matrix mineralization process indicating the success of osteogenesis [57]. Strikingly, we found that, in the absence of other stimulators of osteogenic differentiation, these tested particles induced the expression of both early and late osteogenic markers, promoting an increase in ALP production and the matrix mineralization, respectively. Increased ALP activity may result from the activation of the calcium-sensitive receptor (CaR) by Ca^2+^ ions, which is generally accompanied by upregulation of osteocalcin expression and formation of mineralization nodules, as previously reported in a mouse osteoblastic cell model [58] and observed in cultures of m17.ASC. In addition, Ca^2+^ can also modulate the activation of the ERK1/2 pathway, which leads to phosphate-dependent stimulation of osteopontin (OPN) expression in osteoblasts [59]. Previous findings reported for MC3T3-E1 cells demonstrated that Ca^2+^ released from Ap can enhance osteogenic differentiation by increasing the expression of bone sialoprotein and OPN [60]. These previous studies suggest that Ca^2+^ released from Ap particles would enhance the expression of osteogenesis-related proteins by activating CaR and intracellular signaling pathways that promote osteogenic differentiation of mesenchymal cells.

Concerning the use of both biogenic CaCO_3_ and derived apatites, the use of carbonate minerals from different organisms has been considered a valuable supply for medical applications [61]. The use of calcium phosphates for bone replacement, augmentation, and repair has gained clinical acceptance in many areas of orthopedics and dentistry [62]. The Ap obtained from ES exhibited high biocompatibility and osteoconductive capacity, which, coupled with its architecture, including specific shapes, variable sizes, and porosity, make it suitable as a candidate for preparing materials in bone tissue engineering. Due to its significant potential for conversion of CaCO_3_ into Ap, the mineral content obtained from ES was proven to be as useful as that obtained from other, previously tested biological carbonate hard tissues such as bivalve shell nacreous and coral skeleton aragonite [63,64]. Calcium carbonate obtained from corals can be hydrothermally converted to Ap to mimic the chemical composition of human bone with an added osteoconductive effect [64], which could anticipate the effects achievable through the use of mineral content obtained from ES. It has been reported that coral and non-coral calcium phosphate bioceramics are not only remarkably versatile and clinically safe materials for the human body but also are able to deliver bioactive factors that interact with native bone metabolic cycles [64].

Traditional calcium phosphate-based synthetic materials have several limitations, among which worth highlighting are their poor capacity to induce osteogenesis and their slow resorption, whereas biological minerals have shown better resorption dynamics [54]. The results showed that depending on the OM content, the complete transformation from CaCO_3_ into Ap occurs at 180 or 160 °C, below which the aforementioned transformation remains only partial. The latter feature could be also useful to guarantee gradual or differentiated resorption for replacement with newly synthesized tissue in bone tissue engineering. This behavior has been previously described in corals, in which aragonite was partially transformed into apatite so that an inner core of bioaragonite was covered with an outer shell of apatite. As aragonite is more soluble than apatite, the controlled conversion of bioaragonite into apatite ensures a guided biodegradation rate that favors better bone remodeling and turnover [64]. On the other hand, due to their biocompatibility and low solubility, completely transformed carbonated-apatite particles can be helpful in the manufacture of oral care products for wide-ranging applications, such as remineralization of dental enamel with toothpaste or mouthwashes, reduction of dental hypersensitivity, control of oral biofilm, and dental whitening [65].

## 5. Conclusions

The present research applied the “one-pot” hydrothermal method to transform ES calcite into apatite micro-nanoparticles, presenting biocompatible and osteoinductive features, and analyzed the effect of the OM mainly at the surface level during the mineral replacement reaction. The analyses showed different findings concerning the minimum temperature to achieve complete conversion and the morphology of the Ap crystals when transformed from ES(u) and ES(t) particles. In particular, the transformation in the ES(u) group occurred at a lower temperature, and the Ap crystals mostly displayed hexagonal plate morphologies and hollow crystals. These crystalline formation mechanisms were approached by considering crystallographic aspects and the influence of the surface and intracrystalline OM in the formation of these striking morphologies. Further studies are needed to explore the implications of specific eggshell proteins in the hydrothermal transformation of calcite into apatite. The eggshell particles and apatite micro-nanoparticles were cytocompatible when incubated with MG-63 human osteosarcoma cells and m17.ASC murine mesenchymal stem cells, and were able to promote the osteogenic differentiation of m17.ASC cells. Because of their physicochemical and biological features, these materials can be potentially applied in traumatology for applications in bone replacement, augmentation, and repair, as well as in dentistry for the manufacture of oral care products.

## Figures and Tables

**Figure 1 nanomaterials-13-02299-f001:**
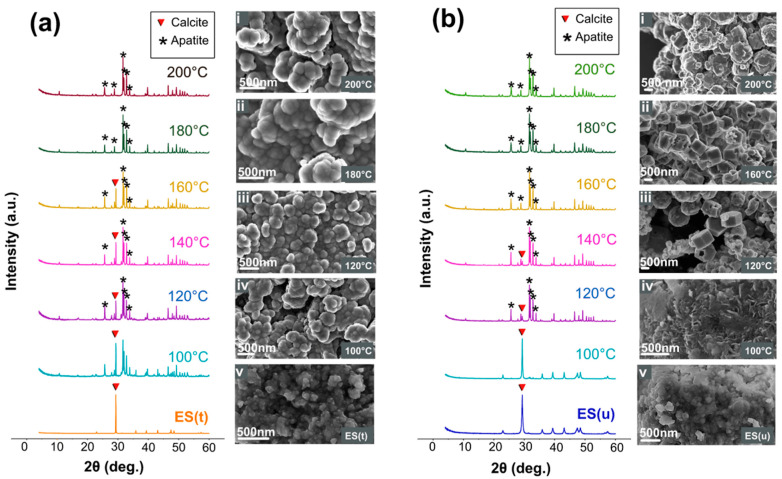
XRD patterns and SEM images of (**a**) ES(t) and (**b**) ES(u) before and after being submitted to hydrothermal conversion between 100 and 200 °C.

**Figure 2 nanomaterials-13-02299-f002:**
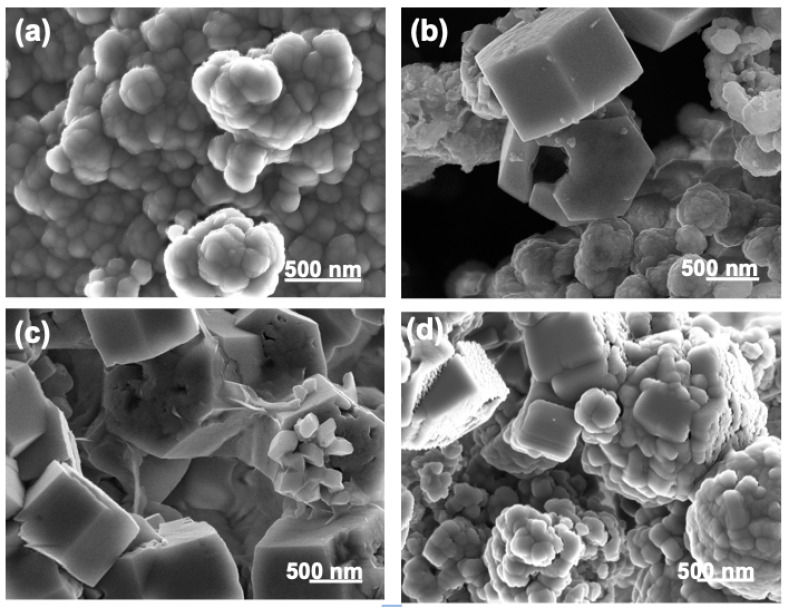
SEM images of (**a**) Ap(t)-180 °C, (**b**) Ap(u)-120 °C, (**c**) Ap(u)-160 °C, and (**d**) Ap(u)-200 °C.

**Figure 3 nanomaterials-13-02299-f003:**
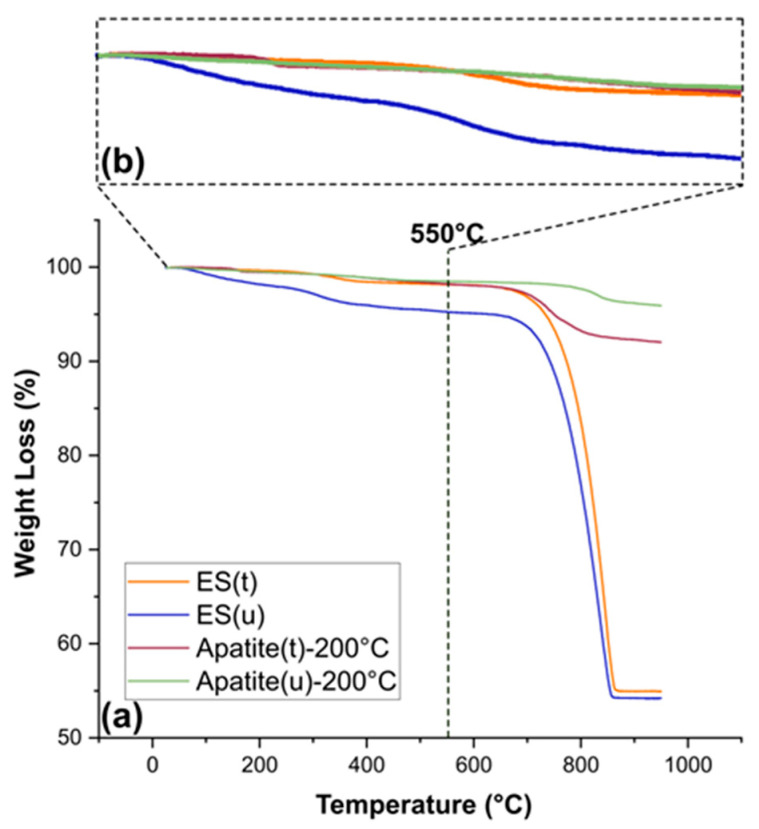
Thermogravimetric analysis (TGA) showing the heating curves for (**a**) ES(u), ES(t), Ap(u)-200 °C, and Ap(t)-200 °C up to 850 °C. (**b**) Idem up to 550 °C. The ranges of temperatures for the calculation of each component were the following: water (up to 180 °C), organic matter (180–550 °C), carbonate mineral (550–850 °C) and total mineral content (>550 °C) content.

**Figure 4 nanomaterials-13-02299-f004:**
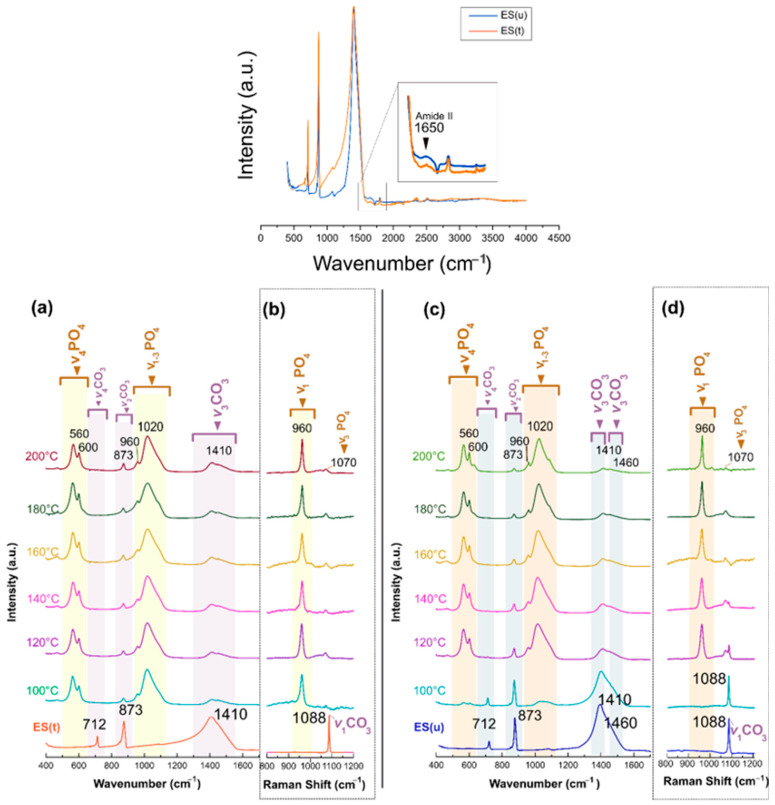
ATR-FTIR and Raman spectra of (**a**,**b**) ES(t) and (**c**,**d**) ES(u) before and after being submitted to hydrothermal conversions between 100 °C and 200 °C.

**Figure 5 nanomaterials-13-02299-f005:**
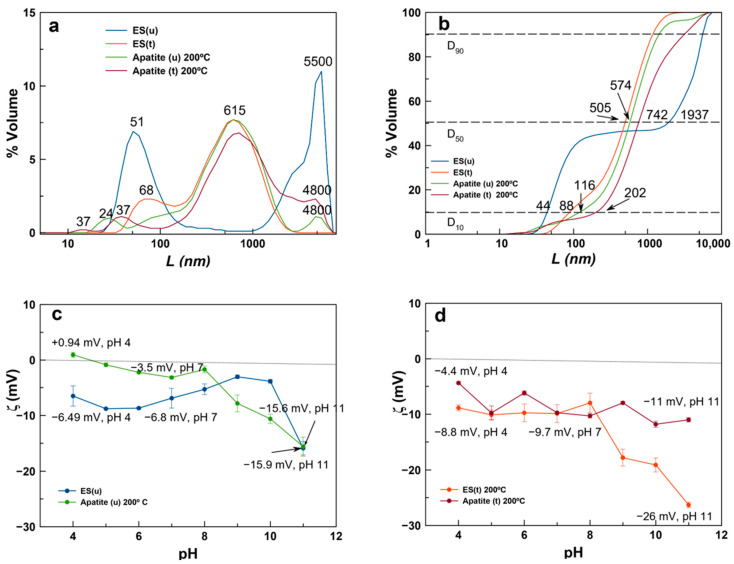
Volume-based CSD (**a**) and cumulative volume-based CSD (**b**) of ES(u), ES(t), Ap(u)-200 °C and Ap(t)-200 °C. ζ-potential versus pH of ES(u) and apatite(u) (**c**) and ES(t) and apatite(t) (**d**). The apatites were prepared by hydrothermal conversion at 200 °C.

**Figure 6 nanomaterials-13-02299-f006:**
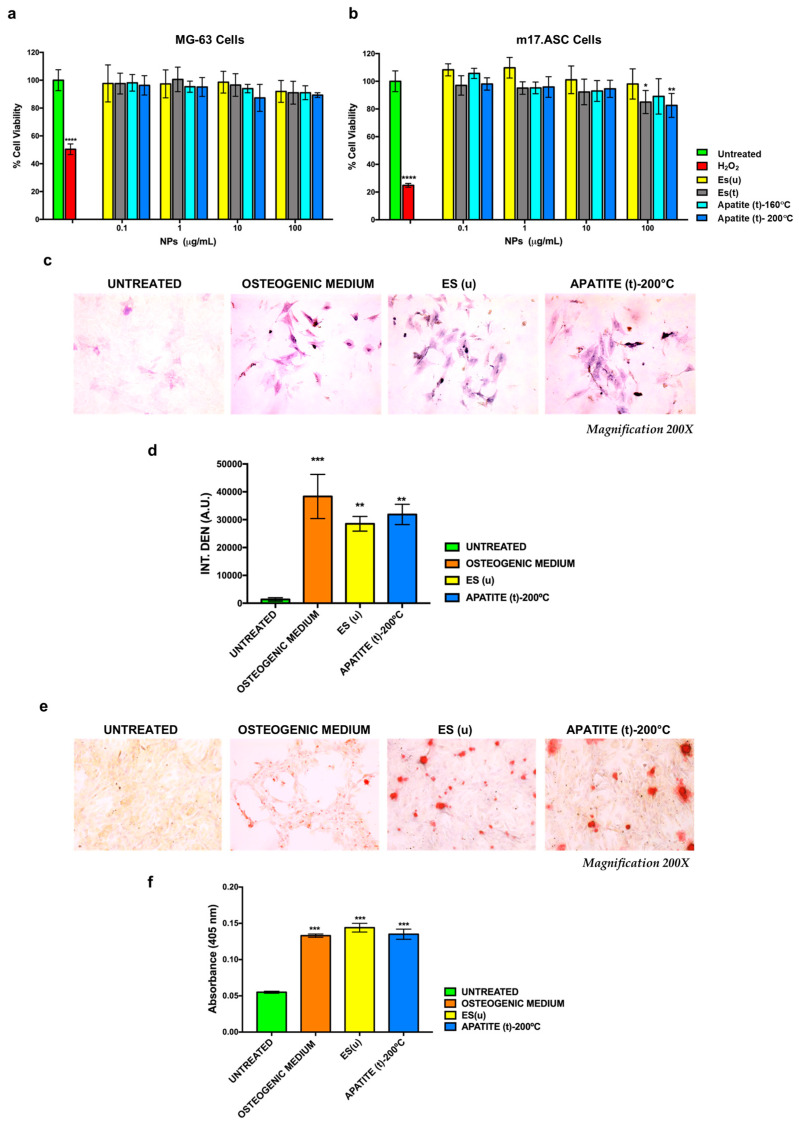
Biological tests results. Viability of MG-63 (**a**) and m17.ASC cells (**b**) incubated for 3 days with different concentrations of particles ES(u), ES(t), Ap(t)-160 °C, and Ap(t)-200 °C. Viability was assessed in MTT assays. Effects of ES(u) and Ap(t)-200 °C particles on ALP activity of m17.ASC cells after 14 days of incubation with 25 μg/mL particles or with the osteogenic induction medium used as positive control. (**c**) Representative images of ALP staining assay obtained with optical microscopy. Magnification 200×. (**d**) Quantitative analysis of ALP activity obtained by counting ALP spots with ImageJ software. Effects of ES(u) and Ap(t)-200 °C particles on the mineralization process of m17.ASC cells. (**e**) The extracellular calcium deposition was visualized by Alizarin Red S staining after cells were cultured with 25 μg/mL particles or with the osteogenic induction medium used as a positive control. Representative images of ARS staining assay obtained with optical microscopy Magnification 200×. (**f**) Mineralization was quantified following the colorimetric analysis of Alizarin Red S solution from calcium deposition (optical density measured in a multi-well reader at 405 nm). Data representing three independent experiments are reported in the histogram as an average (A.U. ± SD), and statistical analyses were carried out using one-way ANOVA with Bonferroni comparison test. Significance was considered as follows: * *p* < 0.05, ** *p* < 0.01, *** *p* < 0.001, **** *p* < 0.0005.

**Table 1 nanomaterials-13-02299-t001:** Chemical composition (wt.%) obtained from TG data analyses. Temperature ranges considered for the calculation of each component: water (up to 180 °C), organic matter (180–550 °C), carbonate mineral (550–850 °C), and total mineral (>550 °C) contents.

	ES(u)	ES(t)	Ap(u)-200 °C	Ap(t)-200 °C
Water	1.64	0.30	0.42	0.52
OM	3.09	1.55	1.07	1.26
Mineral	95.27	98.15	98.51	98.22
Carbonate (550–850 °C)	-	-	1.92	5.65

## Data Availability

Data are contained within the article or Appendix A. Any other data will be shared by the corresponding authors upon request.

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
