# Peer review of "Hydrothermal Transformation of Eggshell Calcium Carbonate into Apatite Micro-Nanoparticles: Cytocompatibility and Osteoinductive Properties"

_nanomaterials, 2023, doi:10.3390/nano13162299_

Round 1

Reviewer 1 Report

It is interesting to fabricate apatite micro-nanoparticles from eggshell calcium carbonate. The studies are systematic. It can be considered after minor corrections.

1. The 8 figures can be combined to 4-5 figures for better reading.

2. The particles were aggregated in the SEM images probably due to the sample preparation process. The size and the dispersion of the particles should be well analyzed.

3. The size distribution of the particles should be quantified.

4. The mechanism of the particle-induced cell osteogenic differentiation should be discussed.

5. Some very related recent papers should be cited and discussed.

Ma et al. Eggshell-derived amorphous calcium phosphate: Synthesis, characterization and bio-functions as bone graft materials in novel 3D osteoblastic spheroids model

Wang et al. Platinum nanoparticles enhance osteogenic differentiation of human dental follicle stem cells via scavenging ROS

Author Response

Manuscript ID: nanomaterials-2537217
Type of manuscript: Article

Date: 05/08/2023

Dear Reviewer,

Please find below the answers to your comments and the actions we have done to update the manuscript accordingly.

It is interesting to fabricate apatite micro-nanoparticles from eggshell calcium carbonate. The studies are systematic. It can be considered after minor corrections.

The authors appreciate the reviewer's comments and suggestions on the manuscript.

  1. The 8 figures can be combined into 4-5 figures for better reading.

The figures related to the biological results (Figures 6-7-8 -> Figure 6) have been combined in order to reduce the final number and improve the readability of the manuscript.

  1. The particles were aggregated in the SEM images probably due to the sample preparation process. The size and the dispersion of the particles should be well analyzed.

The reviewer's comment is correct. The average particle sizes from the SEM images have been obtained for the Ap(u)-200 °C and Ap(t)-200 °C samples.  The following information has been incorporated into the manuscript:

“The average particle sizes for the Ap(u)-200 °C and Ap(t)-200 °C samples measured from SEM images were 87.5 ±13.8 and 131.3 ± 10.6 nm, respectively”

Also in this section (lines 256-257), we highlight that the crystallite size determined by XRD is the crystalline domain size.

  1. The size distribution of the particles should be quantified.

Crystal size distributions (CSD) were calculated using a Malvern Zetasizer Nano ZS analyzer (Malvern Instruments Ltd., Malvern, UK). Volume-based CSD and cumulative volume-based CSD results for ES(u), ES(t), Ap(u)-200 °C, and Ap(t)-200 °C are shown in Figure 5 and described in the manuscript (lines 348-377). The CSD of ES(u) shows particle sizes of around 50 nm and aggregates of up to 5.5 µm, while ES(t) is composed of particles with a median size of 68 nm and aggregates of 615 nm.  The plot as cumulative volume-based distribution (Figure 5b) reveals the percentiles D10, D50, and D90 of the distribution. In these samples, D10 is 44 and 88 nm, respectively, close to the individual particle size, while D50 (the median of the distribution) and D90 are affected by aggregation. On the other side, Ap(u)-200 °C and Ap(t)-200 °C show D10 116 and 202 nm, and D50 574 and 742 nm, respectively, reflecting the presence of short-range aggregates in both samples. In these distributions, D90 is affected by long-range aggregation.

  1. The mechanism of the particle-induced cell osteogenic differentiation should be discussed.

The reviewer's comment is correct. The mechanism of particle-induced osteogenic cell differentiation has been described in more detail in the discussion. The text has been modified accordingly (lines 516-543).

“Particle-induced osteogenic cell differentiation has been linked to the presence of Ca2+ ions, which functions as a potential osteoinductive trigger in mesenchymal cells, as previously established in human bone marrow-derived mesenchymal stromal cells (hBMSC) [54,55]. Ca2+ participates as a second messenger for the activation of the intracellular pathways since Ca-mediated signals couple the membrane events generated by extracellular signals with the cytoplasmic biochemical cascades and nuclear gene expression programs required for cell differentiation. Furthermore, Ca2+ promotes the expression of proteins, including bone morphogenetic protein 2 (BMP-2), which participates in the osteogenic differentiation of hBMSCs [56], which suggests a similar behavior in m17.ASC cells is likely.  In particular, we assessed the ability of both ES (u) and Ap (t)-200 °C to affect the enzymatic ALP activity and the extracellular calcium depositions. ALP plays a critical role during the early stage of osteogenesis and generally reaches the maximum level at two weeks, while extracellular calcium nodules are a unique feature of the matrix mineralization process indicating the success of osteogenesis [57]. Strikingly, we found that, in the absence of other stimulators of osteogenic differentiation, these tested particles induced the expression of both early and late osteogenic markers promoting an increase in ALP production and matrix mineralization, respectively. Increased ALP activity may result from the activation of the calcium-sensitive receptor (CaR) by Ca2+ ions, which is generally accompanied by up-regulation of osteocalcin expression and formation of mineralization nodules, as previously reported in a mouse osteoblastic cell model [58] and observed in cultures of m17.ASC. In addition, Ca2+ can also modulate the activation of the ERK1/2 pathway, which leads to phosphate-dependent stimulation of osteopontin (OPN) expression in osteoblasts [59]. Previous findings reported for MC3T3-E1 cells demonstrated that Ca2+ released from Ap can enhance osteogenic differentiation by increasing the expression of bone sialoprotein and OPN [60]. These previous studies suggest that Ca2+ released from Ap particles would enhance the expression of osteogenesis-related proteins by activating CaR and intracellular signaling pathways that promote osteogenic differentiation of mesenchymal cells.

56. Li, X.; Yang, X.; Liu, X.; He, W.; Huang, Q.; Li, S.; Feng, Q. Calcium carbonate nanoparticles promote osteogenesis compared to adipogenesis in human bone marrow mesenchymal stem cells. Progress in Natural Science: Materials International 2018, 28, 5, 598-608.

57.Barradas, A.; Fernandes, H.; Groen, N.; Chin Chai, Y.; Schrooten, J.; van de Peppel, J, van Leeuwen, J.; van Blitterswijk, C.; de Boer, J. A calcium-induced signaling cascade leading to osteogenic differentiation of human bone marrow-derived mesenchymal stromal cells. Biomaterials 2012, 33, 11, 3205-3215.

58.Aguirre A, González A, Planell JA, Engel E. Extracellular calcium modulates in vitro bone marrow-derived Flk-1 þ CD34þ progenitor cell chemotaxis and differentiation through a calcium-sensing receptor. Biochem Biophys Res Commun 2010, 393, 1, 156e61.

  1. He Y, Zhang H, Teng J, Huang L, Li Y, Sun C. Involvement of the calcium-sensing receptor in inhibition of lipolysis through intracellular cAMP and calcium pathways in human adipocytes. Biochem Biophys Res Commun 2011;404(1):393e9.

60. Saidak Z, Brazier M, Kamel S, Mentaverri Agonists R, Allosteric. Modulators of the Calcium-Sensing Receptor and Their Therapeutic Applications. Mol Pharmacol 2009;76(6):1131e44.

  1. Some very related recent papers should be cited and discussed.

Ma et al. Eggshell-derived amorphous calcium phosphate: Synthesis, characterization, and bio-functions as bone graft materials in novel 3D osteoblastic spheroids model

Wang et al. Platinum nanoparticles enhance osteogenic differentiation of human dental follicle stem cells via scavenging ROS

The authors appreciate the reviewer's recommendation of the references. The following text has been incorporated into the manuscript.

Discussion section:

Lines 506-510. “Other calcium phosphates, such as amorphous calcium phosphate (ACP) particles, have been successfully synthesized from the eggshell through a low-cost guided aggregation technique [52]. These ACP-eggshell derived nanoparticles presented higher resistance to recrystallization, nanostructural control, and biosafety compared to synthetic ACP particles”.

  1. Qianli Ma, Kristaps Rubenis, Ólafur Eysteinn Sigurjónsson, Torben Hildebrand, Therese Standal, Signe Zemjane, Janis Locs, Dagnija Loca, Håvard Jostein Haugen. Eggshell-derived amorphous calcium phosphate: Synthesis, characterization, and bio-functions as bone graft materials in novel 3D osteoblastic spheroids model, Smart Materials in Medicine, (4), 2023, 522-537. 10.1016/j.smaim.2023.04.001.

Introduction section:

Lines 79-83. In bone tissue regeneration, the use of other inorganic nanoparticles, for example, platinum nanoparticles (Pt NPs), has been reported to enhance the osteogenic differentiation of human dental follicle stem cells [21]. However, particles based on Ag NPs present advantages in terms of compositional compatibility with calcified tissues.

  1. Zheng Wang, Jiaxun Wang, Jiacheng Liu, Yating Zhang, Jingyi Zhang, Ruimeng Yang, Zhaosong Meng, Xiaoqun Gong, Lei Sui, Platinum nanoparticles enhance osteogenic differentiation of human dental follicle stem cells via scavenging ROS, Smart Materials in Medicine, 4, 2023, 621-638. 10.1016/j.smaim.2023.06.004.

Reviewer 2 Report

Every year, people eat huge numbers of chicken eggs, the shells of which can be a valuable source of minerals such as apatite. The reviewed article presents the results of research on the transformation of eggshell particles into calcium phosphate, which may make them a potential source of biomedical materials. The introduction contains properly selected information prepared on the basis of new and latest scientific literature. The obtained results of scientific research have been supported by well-designed and conducted analyses, so they do not raise doubts. Apart from unnecessary spaces in the text (lines: 481, 500 and 515), the drawing at the top of page 2 raises the greatest doubt. It has no description. There is no reference to it in the text either. Do the authors treat it as a form of graphical abstract? With these minor shortcomings corrected, I believe the article could be published in the Nanomaterials.

Author Response

Manuscript ID: nanomaterials-2537217
Type of manuscript: Article

Date: 05/08/2023

Dear reviewer, please find below the answer to your comments:

Every year, people eat huge numbers of chicken eggs, shells of which can be a valuable source of minerals such as apatite. The reviewed article presents the results of research on the transformation of eggshell particles into calcium phosphate, which may make them a potential source of biomedical materials. The introduction contains properly selected information prepared on the basis of new and latest scientific literature. The obtained results of scientific research have been supported by well-designed and conducted analyses, so they do not raise doubts. Apart from unnecessary spaces in the text (lines: 481, 500, and 515), the drawing at the top of page 2 raises the greatest doubt. It has no description. There is no reference to it in the text either. Do the authors treat it as a form of graphical abstract? With these minor shortcomings corrected, I believe the article could be published in the Nanomaterials.

The authors appreciate the reviewer's comments on the manuscript. The authors have revised the manuscript for possible editing errors (e.g. blank spaces). Certainly, the figure referred to by the reviewer is the graphical abstract and therefore does not require any specific description.
